# Removal of *Amoxicillin* Antibiotic from Polluted Water by a Magnetic Bionanocomposite Based on Carboxymethyl Tragacanth Gum-*Grafted*-Polyaniline

**Seyedeh Soghra Mosavi [1], Ehsan Nazarzadeh Zare [1,\*] , Hossein Behniafar [1] and Mahmood Tajbakhsh [2]**

[1] School of Chemistry, Damghan University, Damghan 36716-45667, Iran
[2] Department of Organic Chemistry, Faculty of Chemistry, University of Mazandaran, Babolsar 47416-13534, Iran
\* Correspondence: ehsan.nazarzadehzare@gmail.com or e.nazarzadeh@du.ac.ir

**Abstract:** Removal of antibiotics from contaminated water is very important because of their harmful effects on the environment and living organisms. This study describes the preparation of a bionanocomposite of carboxymethyl tragacanth gum-*grafted*-polyaniline and $\gamma Fe_2O_3$ using an in situ copolymerization method as an effective adsorbent for *amoxicillin* antibiotic remediation from polluted water. The prepared materials were characterized by several analyses. The vibrating sample magnetometer and thermal gravimetric analysis showed that the carboxymethyl tragacanth gum-*grafted*-polyaniline@ $\gamma Fe_2O_3$ bionanocomposite has a magnetization saturation of 25 emu g$^{-1}$ and thermal stability with a char yield of 34 wt%, respectively. The specific surface area of bionanocomposite of about 8.0794 m$^2$/g was obtained by a Brunauer–Emmett–Teller analysis. The maximum adsorption capacity (909.09 mg/g) of carboxymethyl tragacanth gum-*grafted*-polyaniline@ $\gamma Fe_2O_3$ was obtained at pH 7, an agitation time of 20 min, a bioadsorbent dose of 0.005 g, and *amoxicillin* initial concentration of 400 mg/L. The Freundlich isotherm and pseudo-second-order kinetic models were a better fit with the experimental data. The kinetic model showed that chemical adsorption is the main mechanism for the adsorption of *amoxicillin* on the bioadsorbent. In addition, the maximum adsorption capacity for *amoxicillin* compared to other reported adsorbents showed that the prepared bionanocomposite has a higher maximum adsorption capacity than other adsorbents. These results show that carboxymethyl tragacanth gum-*grafted*-polyaniline@ $\gamma Fe_2O_3$ would be a favorable bioadsorbent for the remediation of amoxicillin from contaminated water.

**Keywords:** *Amoxicillin* remediation; adsorption; magnetic bionanocomposite; carboxymethyl tragacanth gum; polyaniline; $\gamma Fe_2O_3$

## 1. Introduction

Pharmaceuticals, particularly antibiotics, represent a concerning new class of pollutants in the environment, not to mention their effects on human health with regard to several infectious illnesses [1,2]. Most of these chemicals are found in surface water at elevated amounts [3]. Their bioresistant character and ability to elude traditional sewage treatment procedures are strongly tied together. It harms the ecology by making bacteria resistant and interfering naturally with the growth, development, and movement of a variety of microorganisms [4]. One of the biggest obstacles to a sustainable water future is the presence of this material in the aquatic environment, particularly in arid nations where water recycling is crucial. As a result of their extensive use in both human and animal medications, antibiotics play a significant role in water contamination. They are employed to treat bacterial infections and are extremely resistant to being inactivated until they have completed their intended function, which causes their incomplete metabolism in the organism [5]. More than 90% of medications taken orally don't decompose, thus they become active compounds. Since antibiotics are highly soluble in water, conventional

treatment procedures cannot eliminate them, which poses a significant obstacle to their removal [6].

*Amoxicillin* is a commonly used antibiotic whose structure is based on a β-lactam antibiotic categorized as penicillin, which is the cause of its great bacterial resistance to many microorganisms [7]. Systemic, bacterial, and gastrointestinal illnesses are treated with *amoxicillin* in both human and veterinary medicine. It is widely recognized that *amoxicillin* is utilized in modern medicine, and its ecotoxicity contributes to the danger of medical wastewater. Due to the difficulty of breaking down this antibiotic, the residue is eliminated in the urine and feces. Consuming too much *amoxicillin* creates resistant bacteria because it accumulates in the body and feeds the organisms [8]. Consequently, an effective technique to remove it is essential before it is released into the aquatic environment.

The most common techniques for removing antibiotics include electrochemical degradation [9], Fenton oxidation process [10], UV radiation [11,12], ozonation [13], membrane filtration [14,15], photolysis [16], biological degradation [17], and adsorption [18]. The most appealing technology, nevertheless, is the adsorption process, which has a flexible and straightforward design, is simple to use, is inexpensive, and is highly effective [19,20]. The adsorbent used in industrial applications should be able to quickly absorb the target material and be ecologically benign [21]. The choice of an appropriate adsorbent for antibiotic elimination was the subject of several investigations. In order to effectively remove *amoxicillin* from water, a range of micro/nanostructures were deemed acceptable adsorbents, either as single phases or composites. Several works have reported the use of various adsorbents for the removal of *amoxicillin* from aqueous solutions. For example, a metal–organic framework (MIL-53(Al)) was prepared with a hydrothermal technique and used as an adsorbent for the removal of *amoxicillin* antibiotics from water [22]. It was reported that the adsorption capacity of MIL-53 is 758.5 mg $g^{-1}$ in experimental conditions due to its high surface area. In another work, an adsorbent based on $NH_4Cl$-induced activated carbon was employed for the removal of *amoxicillin* from water [23]. It was expressed that the removal percentage of *amoxicillin* is above 99%, owing to the high specific surface area (1029 $m^2/g$) of $NH_4Cl$-induced activated carbon. *Amoxicillin* removal from water was reported by an adsorbent based on activated carbon produced from pomegranate peel/iron nanoparticles [24]. The high removal percentage (97.9%) was obtained at a pH of 5 during a contact time of 30 min. A green magnetic adsorbent based on functional $CoFe_2O_4$-modified biochar was used for *amoxicillin* removal from water. It was reported that the maximum adsorption capacity (99.99 mg/g) is obtained at a pH of 7 and at ambient temperature [25].

Of those, given their biodegradability, availability, reasonable cost, and minimal toxicity to biological systems, biopolymer-based composites have recently been used often as an ecologically friendly adsorbent to remove water contaminants. Natural polymer hydrogels were created using guar gum, karaya gum, xanthan gum, ghatti gum, and tragacanth gum. These natural polymers were capable of efficiently capturing pollutants in their three-dimensional (3D) network. As a result, the amount of interaction between contaminants and the hydrogel's surface functional groups is increased, improving adsorption effectiveness. They also have several industrial uses because of their capacity to hydrate in cold or hot water, either by stabilizing emulsion systems or by gel formation.

Tragacanth gum (TG), as a colorless and odorless natural polymer, is a highly complex heterogeneous anionic polysaccharide that forms from the stems and branches of *Astragalus gummier* and other Asian *Astragalus* species [26]. Within a few weeks, the exudate can be recovered after it has solidified into flakes or coils of ribbon [27]. Tragacanth gum is a substance that is found in Iran, India, Afghanistan, and Turkey. Neutral and anionic sugars, such as D-galacturonic acid, D-galactose, D-xylose, L-arabinose, L-fucose, and d-glucose are found in TG [26,28]. This natural polymer is widely used in the food industry, in medicine, and in cosmetics [29]. In addition, it is inexpensive, readily accessible, and has great solubility, strong thermal stability, and a long shelf life [30]. The existence of the hydroxyl- and carboxylic-acid-reactive functional groups in the TG can be used as chelating sites for the removal of pollutants from water. One method for improving the adsorption

capacity of natural polymers is copolymerization with functional monomers and adding inorganic fillers.

The aforementioned advantages can be strengthened by developing bionanocomposite materials that contain polyaniline and magnetic nanoparticles, such as $\gamma Fe_2O_3$. Consequently, in this study, a three-step oxidative polymerization procedure using ammonium persulfate oxidizer was used to create an antibacterial magnetic bionanocomposite based on carboxymethyl tragacanth gum-*grafted*-polyaniline, and then various aspects of it were examined. The bionanocomposite was subsequently applied for *amoxicillin* antibiotic removal from polluted water.

## 2. Materials and Methods

### 2.1. Materials and Instruments

Distilled aniline, ammonium persulfate, monochloroacetic acid, hydrochloric acid, iron(III) chloride hexahydrate, iron(II) chloride tetrahydrate, and other solvents were supplied by Merck Company (Darmstadt, Germany). Tragacanth gum (TG) with high-quality in translucent flakes was purchased from Rad Kimia-Garan Company (Tehran, Iran).

The chemical structure of prepared materials was studied by Fourier transform infrared (FTIR, Equinox 55, Bruker, Leipzig, Germany), elemental analysis (CHNSO, ECS 4010, NC technologies) and energy-dispersive X-ray (EDX, MIRA 3-XMU, Tescan, Kohoutovice, Brno-Kohoutovice Czech Republic). The crystallinity and morphology of the samples were assessed by X-ray diffraction (XRD, D8 Advance X-ray diffractometer, Bruker, Leipzig, Germany) and a field emission scanning electron microscope (FESEM, MIRA 3-XMU, Tescan, Kohoutovice, Brno-Kohoutovice Czech Republic). The specific surface area of the samples was studied by the Brunauer–Emmett–Teller (BET, Belsorp mini II, MicrotracBEL, Osaka, Japan) technique. The thermal behavior of the samples was investigated by thermogravimetric analysis (TGA, L81A1750, Linseis, Selb, Germany).

### 2.2. Preparation of Carboxymethyl Tragacanth Gum (CMT)

The carboxymethyl tragacanth gum (CMT) was synthesized according to our previous literature with slight modifications [31]. We fully dissolved 1 g of powdered tragacanth gum (TG) in a mixture of water/ethanol (85 mL/15 mL). Then, 1.2 g of NaOH in 10 mL of distilled water was added to the reaction mixture. The solution was kept at 50 °C for 30 min under a magnetic stirrer (Heidolph, Schwabach, Germany). Afterward, 1.3 g of monochloroacetic acid in 10 mL of distilled water was added to the reaction solution, and the final solution was stirred for 4 h at 50 °C. After cooling, the solution was poured into a double volume of ethanol or methanol. The resultant CMC sediment was separated using filter paper and dried at 40 °C (Figure 1).

### 2.3. Synthesis of $\gamma Fe_2O_3$

The co-precipitation technique was employed for the $\gamma$-$Fe_2O_3$ nanoparticles synthesis [32]. A solution of NaOH (1.0 M) was added to 100 mL of deionized water, and the reaction mixture was stirred magnetically for 15 min under an inert atmosphere. Afterward, a solution of iron(III) and iron(II) salts was dropped into the previous solution, and the final solution was kept under vigorous stirring at ambient temperature for 70 min. The resulting brown precipitate was separated and washed several times with distilled water and ethanol. Lastly, the precipitate was dried and the obtained powder was calcined at 300 °C for 2 h to gain the $\gamma$-$Fe_2O_3$ nanoparticles.

**Figure 1.** Schematic illustration of the preparation of carboxymethyl tragacanth gum (CMT) and carboxymethyl tragacanth gum-*grafted*-polyaniline@$\gamma Fe_2O_3$ (CMT-g-PANI@Fe$_2$O$_3$) bionanocompsite.

## 2.4. Preparation of Carboxymethyl Tragacanth Gum-Grafted-Polyaniline@$\gamma Fe_2O_3$

In a 250 mL round-bottom flask, 0.62 g of carboxymethyl tragacanth gum (CMT) was dissolved in 50 mL of distilled water. Then, 1.5 mL of HCl was added to the solution, and the flask was kept under $N_2$ gas at 0–5 °C. Subsequently, 10 wt% (an optimal amount) of $\gamma Fe_2O_3$ nanoparticles were dispersed in 20 mL of distilled water for 30 min using an ultrasonic bath and then added to the above solution. After that, within 30 min, 3 g of ammonium persulfate in 10 mL of distilled water was added to the solution. The flask was then placed on a magnetic stirrer for 12 h under the aforementioned conditions after 1.25 mL of aniline monomer was added. The nanocomposite was separated by a magnet and rinsed with 10 mL of *N*-methyl pyrrolidone, water, and acetone before they were dried at 50 °C (Figure 1).

## 2.5. Adsorption Experiment

The carboxymethyl tragacanth gum-*grafted*-polyaniline@$\gamma Fe_2O_3$ (CMT-g-PANI@Fe$_2$O$_3$) bionanocomposite's ability to remove *amoxicillin* antibiotic from aqueous solutions was tested in a few different ways. The calibration curve of *amoxicillin* was prepared by defining the absorbance at 228 nm with a series of standard solutions (1−8 mg/L) achieved from reducing the stock solution at pH 7. Then, the *amoxicillin* initial or equilibrium concentrations were measured with the calibration curve [22].

*Amoxicillin's* initial concentration in an aqueous solution, solution pH, the amount of adsorbent, the agitation time, and other important parameters were all examined for their effects on adsorption capacity. The pH was changed from 4 to 9 using HCl (0.1 N) and NaOH (0.1 N). The optimal adsorption conditions were then investigated using a range of CMT-g-PANI@Fe$_2$O$_3$ biosorbent doses (0.005–0025 g), contact periods (5–30 min), and initial *amoxicillin* concentrations (50–400 ppm). By comparing the outcomes of the experimental data with those predicted by the Freundlich and Langmuir models, the adsorption isotherms were also explored. To assess the adsorption kinetics, the pseudo-first-order and pseudo-second-order models were also applied. The experimental tests were carried out three times, and an average of the results was provided. A UV-visible spectrometer was used to assess the *amoxicillin* concentration. *Amoxicillin's* capacity and adsorption efficiency onto CMT-g-PANI@Fe$_2$O$_3$ bionanocomposite were calculated using Equations (1) and (2), respectively [20].

$$R\% = \left( \frac{C_i - C_e}{C_i} \right) \times 100 \tag{1}$$

$$Q_e = \left(\frac{C_i - C_e}{m}\right) \times V \tag{2}$$

where $C_i$ and $C_e$ are the *amoxicillin* initial and the equilibrium concentrations in the solutions (mg/L), correspondingly. m is the weight (g) of CMT-g-PANI@$Fe_2O_3$ bionanocomposite and V is the solution volume (L).

### 2.6. Isotherm Study

Langmuir and Freundlich's isotherm models are used to evaluate the maximum adsorption capacity and equilibrium adsorption isotherms. The Langmuir isotherm model measured the single-layer adsorption of contaminants onto the adsorbent surface, whereas the Freundlich isotherm model measured the multilayer adsorption of pollutants on the adsorbent surface. The mathematical expression of Langmuir's (Equation (3)) and Freundlich's (Equation (4)) isotherm models are as follows [22,33]:

$$\frac{C_e}{Q_e} = \frac{1}{K_L Q_{max}} + \frac{1}{Q_{max}} C_e \tag{3}$$

$$LnQ_e = LnK_F + \frac{1}{n} LnC_e \tag{4}$$

where, $C_e$, $Q_e$, and $Q_{max}$ are the equilibrium concentration (mg/L), the equilibrium, and maximum adsorption capacity (mg/g) correspondingly; $K_L$ (L/mg) is the Langmuir constant calculated from the plot between $C_e/Q_e$ and $C_e$; $K_F$ (L/mg) is the Freundlich constant calculated from the plot between $Ln\,Q_e$ and $Ln\,C_e$. n is a parameter to define the adsorption process favorability; once n > 1, the *amoxicillin* adsorption onto bioadsorbent is anticipated at high concentrations.

### 2.7. Kinetic Study

The famous kinetics models were used for studying the effect of time on the adsorption process. The mathematical expression of pseudo-first-order (PFO, Equation (5)) and pseudo-second-order (PSO, Equation (6)) models are shown as follows [22,33]:

$$Ln(Q_e - Q_t) = LnQ_e - k_1 t \tag{5}$$

$$\frac{t}{Q_t} = \frac{1}{k_2 Q_e^2} + \frac{1}{Q_e} t \tag{6}$$

where $Q_t$ (mg/g), and $Q_e$ (mg/g) are the adsorption capacity at time t and equilibrium, correspondingly. $k_1$ (1/min) and $k_2$ (g/mg·min) are the rate constants of the PFO and PSO, correspondingly.

### 2.8. Desorption and Reusability

To examine the desorption and reusability of the CMT-g-PANI@$Fe_2O_3$ bionanocomposite, the *amoxicillin* adsorbed onto CMT-g-PANI@$Fe_2O_3$ bionanocomposite was floated in ethanol and stirred at ambient temperature for 1 h. After that, the bionanocomposite was separated by a magnet. The quantity of released *amoxicillin* in the elution medium was measured afterward employing a UV-visible spectrophotometer. The following equation was employed to obtain the desorption percentage.

$$\%D = \frac{A}{B} \times 100 \tag{7}$$

where A is the *amoxicillin* desorbed (mg) in the medium and B is the *amoxicillin* adsorbed (mg) on the CMT-g-PANI@$Fe_2O_3$ bionanocomposite.

## 3. Results and Discussion

Currently, the removal of antibiotic drugs from polluted water due to their destructive effect on humans and other living beings is very important. In this regard, we employed an antibacterial magnetic nanobiosorbent based on a carboxymethyl tragacanth gum-grafted-polyaniline and $\gamma Fe_2O_3$ nanoparticles (CMT-g-PANI@Fe$_2$O$_3$) for the elimination of *amoxicillin* from contaminated water.

### 3.1. Nanobiosorbent Characterization

FTIR spectra are employed for studying the chemical structure of prepared materials and are shown in Figure 2A. In the FTIR spectrum of the TG, the absorption bands at 3470 cm$^{-1}$, 2934 cm$^{-1}$, and 1740 cm$^{-1}$ are related to the stretching vibrations of the OH, –CH, and ester carbonyl groups, respectively [34,35]. In addition, the absorption band at 1604 cm$^{-1}$ is due to the existence of the carboxylate anion of d-galacturonic acid. In the FTIR spectrum of the CMT, the absorption band at 1622 cm$^{-1}$ is associated with the –COO$^-$ asymmetric vibrations which overlapped with the carbonyl groups of acidic and ester existing in the TG parts of the CMT [31,34]. Moreover, the absorption band that appeared at 1433 cm$^{-1}$ is related to the bending vibrations of –CH$_2$. The FTIR spectrum of $\gamma Fe_2O_3$ nanoparticles displayed an absorption band at 550 cm$^{-1}$ that was related to the stretching vibrations of Fe–O. Moreover, the bands at 1620 cm$^{-1}$ and 3400 cm$^{-1}$ were associated with the –OH stretching vibrations on the surface of the $\gamma Fe_2O_3$ nanoparticles [36,37]. The FTIR spectrum of CMT-g-PANI displayed a broad absorption band at ~3205 cm$^{-1}$ and is related to the stretching vibrations of the N–H and OH of PANI and CMT, respectively. The absorption bands at 3050 cm$^{-1}$ and 2949 cm$^{-1}$ are attributed to aromatic and aliphatic C–H in the PANI and CMT, respectively. The two bands appearing at 622 cm$^{-1}$ and 1470 cm$^{-1}$ were related to the asymmetric vibrations of the –COO$^-$ and stretching vibration of the benzenoid ring, respectively [38]. The presence of characteristic absorption bands of CMT, PANI, and Fe$_2$O$_3$ in the FTIR spectrum of CMT-g-PANI@Fe$_2$O$_3$ revealed that the bioadsorbent was synthesized.

The XRD analysis was employed for the investigation of the crystallinity nature of the samples as shown in Figure 2B. The XRD pattern of CMT showed an amorphous nature as compared to the XRD pattern of TG [31,39]. This crystallinity reduction may be related to the the –OH group's replacement by the –COO$^-$ groups in the CMT [34]. A crystalline nature with diffraction peaks located at 2theta = 23°, 27°, 34°, 35°, 41°, 49°, 53°, 57°, 62°, and 63° was observed in the XRD pattern of Fe$_2$O$_3$ [39]. Synthesized Fe$_2$O$_3$ has an ordered cubic structure, which is consistent with JCPDS file no. 39-1346 [40]. XRD patterns comparison of CMT-g-PANI and CMT-g-PANI@Fe$_2$O$_3$ showed that the crystallinity of CMT-g-PANI improved in the presence of Fe$_2$O$_3$.

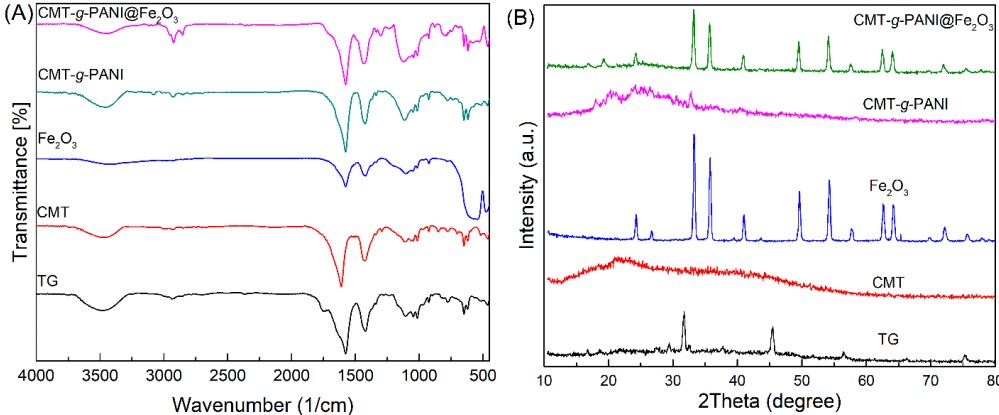

**Figure 2.** FTIR spectra (**A**) and XRD patterns (**B**) of TG, CMT, Fe$_2$O$_3$, CMT-g-PANI, and CMT-g-PANI@Fe$_2$O$_3$.

The elemental analysis of TG, CMT, and CMT-g-PANI was performed, and the percentage mass contents of the elements are shown in Table 1. In CMT, the oxygen content reaches 66.27% as compared to TG which means that the substitution of $-COO^-$ was carried out in the TG, successfully. Correspondingly, in CMT-g-PANI, the oxygen content decreased to 47.71%. This could be related to the grafting reaction between CMT and PANI. Furthermore, the presence of nitrogen content of 8.03% in CMT-g-PANI is related to the PANI.

EDX examination was also applied for the chemical structure study of the prepared samples as shown in Figure 3A. The existence of C, O, Cl, and Na elements in the EDX spectrum of CMT confirmed the preparation of CMT and the substitution of $-COONa$ in the TG backbone. The existence of a tiny amount of Cl is related to chlorine removal from monochloroacetic acid due to the nucleophilic substitution reaction between the hydroxyl of TG and monochloroacetic acid. The presence of Fe and O elements in $Fe_2O_3$ and C, N, O, Cl, and Na elements in the CMT-g-PANI expressed that the $Fe_2O_3$ and CMT-g-PANI@$Fe_2O_3$ were prepared. In addition, the presence of C, O, N, Fe, Cl, and Na in CMT-g-PANI@$Fe_2O_3$ showed that the bionancomposite was prepared successfully.

**Table 1.** Elemental analysis of TG, CMT, and CMT-g-PANI.

| Sample | C | H | N | O |
|---|---|---|---|---|
| TG | 42.19 | 5.71 | - | 52.1 |
| CMT | 29.66 | 4.07 | - | 66.27 |
| CMT-g-PANI | 39.88 | 4.38 | 8.03 | 47.71 |

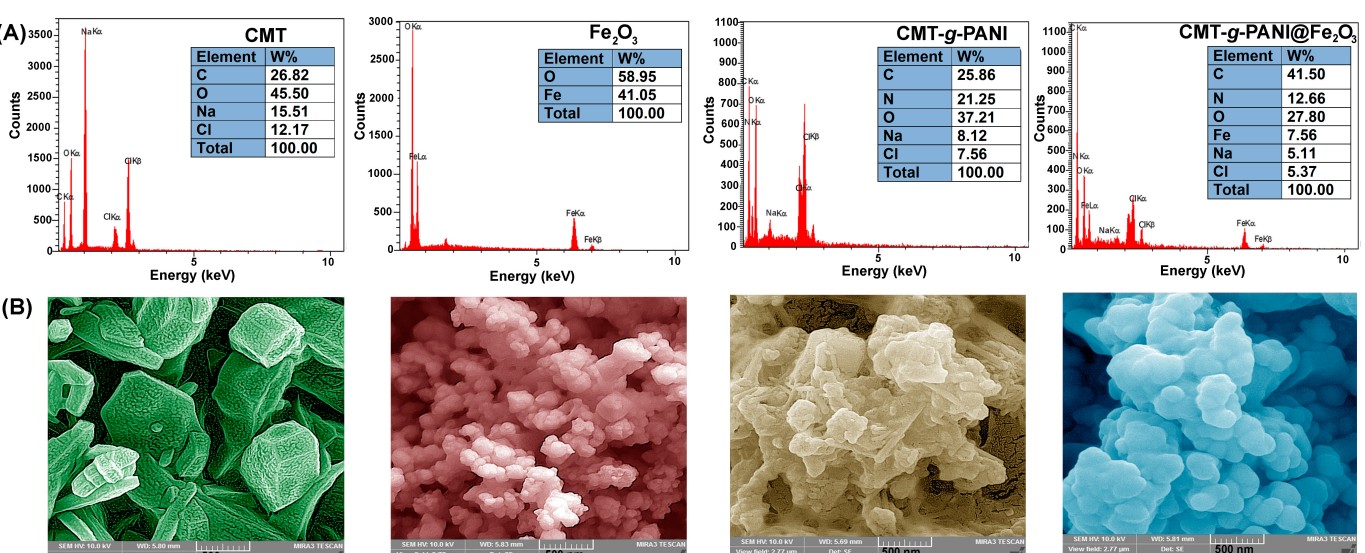

**Figure 3.** EDX spectra (**A**) and FESEM micrographs (**B**) of CMT, $Fe_2O_3$, CMT-g-PANI, and CMT-g-PANI@$Fe_2O_3$.

Figure 3B shows the FESEM micrographs of CMT, $Fe_2O_3$, CMT-g-PANI, and CMT-g-PANI@$Fe_2O_3$. The FESEM image of CMT shows polyhedral particles with microsize. It appears that the chemical modification of TG leads to the formation of polyhedral microparticles. The nano-spherical and amorphous structures were observed in the FESEM images of $Fe_2O_3$ and CMT-g-PANI. The presence of $Fe_2O_3$ nanoparticles lead to a granular structure in the FESEM image of CMT-g-PANI@$Fe_2O_3$.

VSM analysis was employed to evaluate of magnetic properties of $Fe_2O_3$ and CMT-g-PANI@$Fe_2O_3$ as shown in Figure 4A. The magnetization saturation (Ms) values of $Fe_2O_3$ and CMT-g-PANI@$Fe_2O_3$ were 70 emu g$^{-1}$ and 25 emu g$^{-1}$, respectively, and showed

superparamagnetic properties for both. In addition, the Ms value of $Fe_2O_3$ decreased with the coating of $Fe_2O_3$ by CMT-g-PANI.

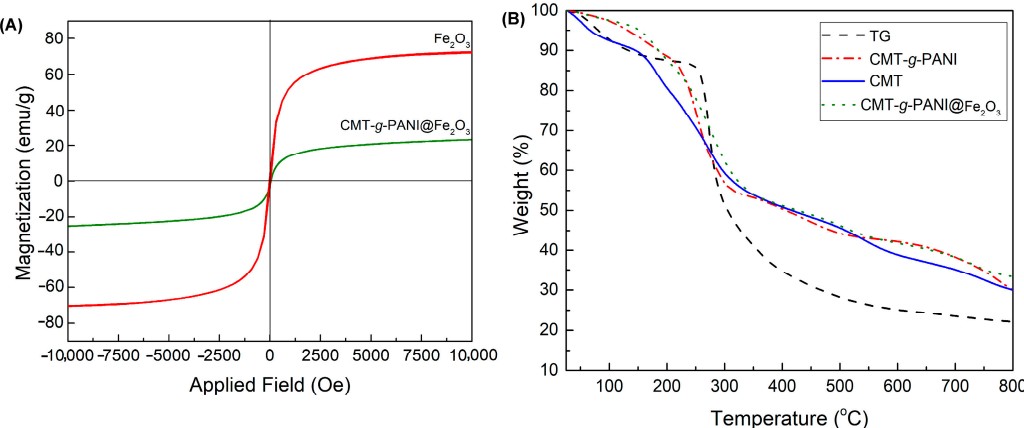

**Figure 4.** VSM curves of $Fe_2O_3$, and CMT-g-PANI@$Fe_2O_3$ (**A**) and TGA thermograms (**B**) of TG, CMT, CMT-g-PANI, and CMT-g-PANI@$Fe_2O_3$.

TG analysis was applied to study the thermal stability of prepared materials as seen in Figure 4B. The TGA thermogram of TG displays two steps of mass loss from 50 to 178 °C and from 200 to 580 °C. The first mass loss (10 wt%) was related to the removal of moisture absorbed in the TG, and the second mass loss (65 wt%) corresponded to the branched heterogeneous structure degradation of the TG [34,39]. The char yield of TG at 800 °C was 25 wt%. Four mass losses at 30–150 °C, 150–300 °C, 300–550 °C, and 550–780 °C were observed in the TGA thermogram of CMT. The first (10 wt%) and second (32 wt%) mass losses were related to the moisture evaporation and combination of the saccharide ring degradation, the C–O–C breaking, and the $CO_2$ elimination from the CMC, respectively [34,39]. The two latter mass losses were attributed to the CMT backbone breaking [39]. The char yield of CMC at 800 °C was 33 wt%. In the TGA thermogram of CMT-g-PANI, four mass losses were observed. The first mass loss (10 wt%) at 5–200 °C was caused by the evaporation of water and solvent entrapment in the copolymer chains [41]. The second mass loss (25 wt%) at 200–300 °C probably corresponded to the loss of hydrochloric acid, saccharide ring, and the $CO_2$ elimination from the CMT-g-PANI. The third mass loss (10 wt%) at 300–500 °C and the fourth mass loss (20 wt%) at 500–750 °C was ascribed to the breakdown of sugar units in the CMT structure and PANI, respectively. The char yield of CMT-g-PANI at 800 °C was 34 wt%. Compared with CMT-g-PANI, the thermogram of the CMT-g-PANI@$Fe_2O_3$ showed good thermal stability owing to the existence of $Fe_2O_3$ nanoparticles in the CMT-g-PANI matrix.

Brunauer–Emmett–Teller (BET) analysis was applied to define the specific surface area of materials and evaluate the effect of the presence of $Fe_2O_3$ nanoparticles within the CMT-g-PANI matrix, as seen in Figure 5. The specific surface area ($a_{s,BET}$) values of CMT-g-PANI@$Fe_2O_3$ and $Fe_2O_3$ were 8.0794 m$^2$/g and 2.8278 m$^2$/g, respectively. This confirms that the CMT-g-PANI@$Fe_2O_3$ had better surface properties in terms of surface area and micropore area compared to $Fe_2O_3$. Thus, the presence of $Fe_2O_3$ together with CMT-g-PANI led to an improvement in the $a_{s,BET}$ value of CMT-g-PANI@$Fe_2O_3$ which could be effective in antibiotic adsorption by bioadsorbent.

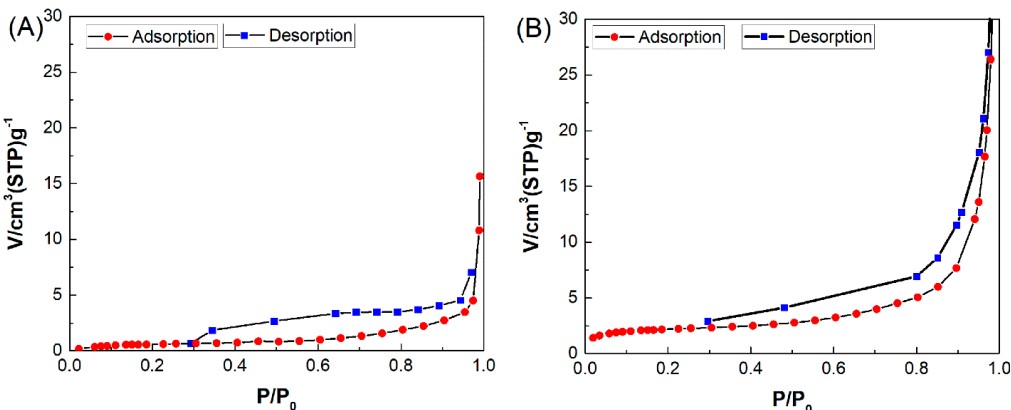

**Figure 5.** $N_2$ adsorption/desorption isotherms of $Fe_2O_3$ (**A**) and CMT-g-PANI@$Fe_2O_3$ (**B**).

### 3.2. Optimization of Effective Parameters for Amoxicillin Adsorption

Solution pH: It is well known that the changes in the medium pH can directly affect the removal percentage of pollutants from an aqueous solution. In this regard, a solution pH range of 4–9 for finding the best pH for *amoxicillin* adsorption by the CMT-g-PANI@$Fe_2O_3$ was examined (Figure 6A). Results revealed that by growing the pH values from 4 to 7, the $Q_e$ increased from 54.18 mg/g to 65.39 mg/g and then decreased at high pH (8 and 9). At pH 7, *amoxicillin* antibiotic occurs as zwitterions, while at pH 4 and 5, it performs in cationic species, and at pH 8 and 9, it appears in anionic species. Thus, *amoxicillin* removal was efficient under pH 7 (neutral). The hydroxyl, carboxyl, and amine groups in the copolymer together with a high surface-to-volume ratio of $Fe_2O_3$ in the CMT-g-PANI@$Fe_2O_3$ adsorbent lead to the rapid diffusion of *amoxicillin* molecules into the adsorbent and interaction with its functional groups for efficient *amoxicillin*.

Amount of biosorbent: Adsorption examinates were carried out with changing quantities (0.005–0.025 g) of the CMT-g-PANI@$Fe_2O_3$ at pH 7 (optimal) to investigate the association between the adsorbent amount and its adsorption capability for *amoxicillin*. According to Figure 6B, the $Q_e$ decreased from 189.73 to 34.50 mg/g as the adsorbent amount increased from 0.005 g to 0.025 g. At a lower adsorbent amount, the higher the amount of *amoxicillin* adsorbed by the CMT-g-PANI@$Fe_2O_3$. The optimal quantity of CMT-g-PANI@$Fe_2O_3$. for subsequent examinations was found to be 0.005 g.

Agitation time: The agitation time effect on the $Q_e$ of the CMT-g-PANI@$Fe_2O_3$ adsorbent for the removal of *amoxicillin* was examined (Figure 6C). Results showed that the $Q_e$ increased up to 195.49 mg/g by increasing the agitation time from 5 to 20 min at pH 7 with an adsorbent dose of 0.005 g (optimal). Furthermore, the $Q_e$ decreased slightly to 183.93 mg/g once the agitation time reached 30 min. Consequently, an agitation time of 20 min was chosen to be the perfect time for subsequent examinations. Generally, at the beginning of the adsorption process, numerous vacant sites of adsorbent are vacant to interact with *amoxicillin* molecules in solution. The functional group's interaction of the adsorbent and the *amoxicillin* grow stronger once the agitation time is increased to 20 min. At longer time frames (>20 min), there was no more improvement in $Q_e$, which was probably related to the existence of adsorbent active sites and the equilibrium state approach.

*Amoxicillin* initial concentration: The association between the *amoxicillin* initial concentration and the $Q_e$ of the CMT-g-PANI@$Fe_2O_3$ adsorbent was studied by altering the *amoxicillin* concentration from 50 to 400 mg/L at the optimal pH (7.0), dose (0.005 g), and agitation time (20 min). Figure 6D shows that the *amoxicillin* initial concentration influences the $Q_e$ of CMT-g-PANI@$Fe_2O_3$ adsorbent. With increasing the *amoxicillin* initial concentration from 50 to 400 mg/L, $Q_e$ of adsorbent with an intensity rose to 788.83 mg/g with a steep slope. Thus, the $Q_e$ increases once the *amoxicillin* initial concentration is increased, whereas the adsorbent amount remains constant, and this pattern holds until the *amoxicillin* initial concentration reaches an equilibrium level.

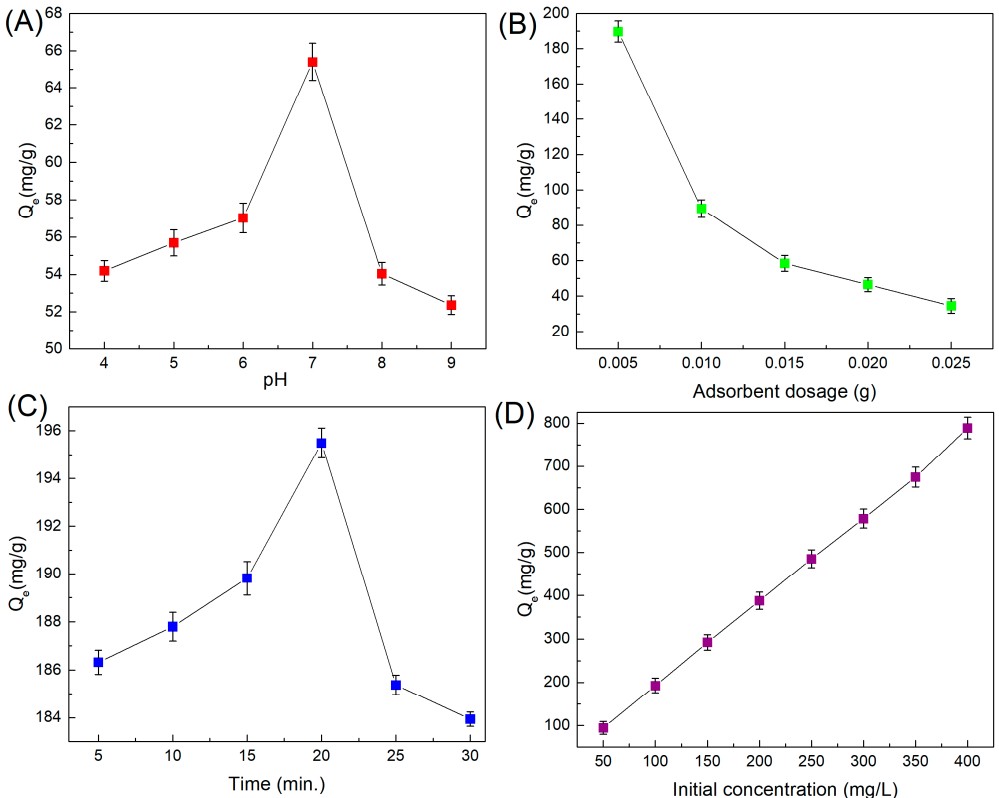

**Figure 6.** (**A**) Effect of pH (4–9), (biosorbent dose = 0.015 g, amoxicillin concentration = 100 mg/L, time = 15 min and temperature = 298 K), (**B**) biosorbent dosage (0.005–0.025 g), (pH 7, *amoxicillin* concentration = 100 mg/L, time = 15 min and temperature = 298 K), (**C**) time (5–30 min), (pH 7, biosorbent dosage = 0.005 g, *amoxicillin* concentration = 100 mg/L, V = 10 mL, and temperature = 298 K), (**D**) *amoxicillin* concentration (50–400 mg/L), (pH 7, biosorbent dosage = 0.005 g, V = 10 mL, time = 20 min, temperature = 298 K).

### 3.3. Adsorption Isotherm

To find the interaction between amoxicillin and CMT-g-PANI@Fe$_2$O$_3$ bioadsorbent, the isotherm study was employed. It is well known that Langmuir and Freundlich's isotherm models are used to evaluate the maximum adsorption capacity and equilibrium adsorption isotherms. Figure 7A,B demonstrates the Langmuir and Freundlich isotherm. The obtained parameters of isotherm models are tabulated in Table 2. Consistent with obtained correlation coefficient (R$^2$) of isotherm models, the Freundlich isotherm was found to be more reliable with the experimental data than the Langmuir model. This result shows that the *amoxicillin* adsorbed over the CMT-g-PANI@Fe$_2$O$_3$ bioadsorbent surface as a multilayer. In addition, the obtained "n" in the Freundlich model is an important parameter to define the adsorption process favorability. The value of n > 1 in Table 2 shows that the *amoxicillin* adsorbed over the CMT-g-PANI@Fe$_2$O$_3$ bioadsorbent surface is desirable at high concentrations.

The comparison of Q$_{max}$ for *amoxicillin* to other adsorbents reported in recent years showed that the CMT-g-PANI@Fe$_2$O$_3$ biosorbent has a higher maximum adsorption capacity (909.09 mg·g) than other adsorbents (Table 3). This could be related to the presence of Fe$_2$O$_3$ nanoparticles and numerous active sites, for example, hydroxyl, carboxylate, and amine groups in the nanocomposite, which can effectively interact with *amoxicillin* (via electrostatic interaction and hydrogen bonding), led to eliminating the *amoxicillin*.

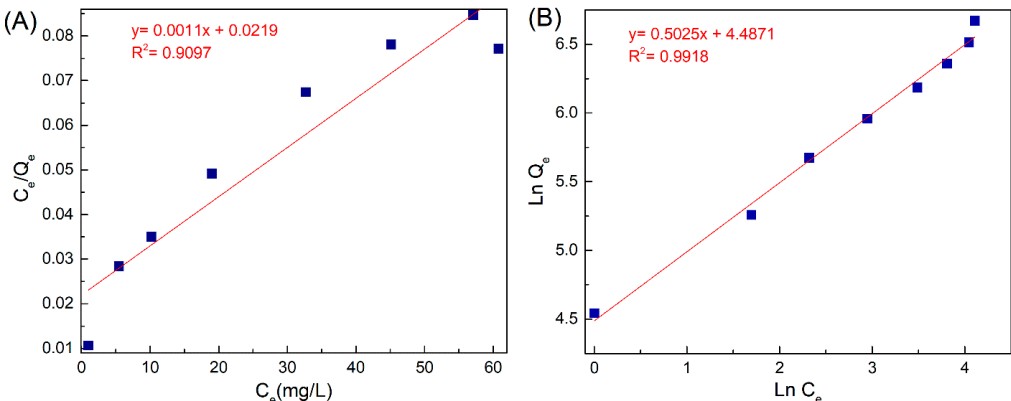

**Figure 7.** (**A**) Langmuir and (**B**) Freundlich isotherms (condition: *amoxicillin* concentration (50–400 mg/L), pH 7, biosorbent dosage = 0.005 g, contact time = 20 min, T = 298 K).

**Table 2.** Isotherm parameters, for amoxicillin adsorption onto the CMT-g-PANI@Fe$_2$O$_3$ bioadsorbent.

| Isotherm | Parameters | |
|---|---|---|
| Freundlich | $K_F$ | 88.8633 |
| | n | 1.99 |
| | $R^2$ | 0.9918 |
| Langmuir | $Q_{max}$ | 909.09 |
| | $K_L$ | 0.0502 |
| | $R^2$ | 0.9097 |

**Table 3.** Assessment of the maximum adsorption capacity of CMT-g-PANI@Fe$_2$O$_3$ bionanocomposite with other reported studies.

| Adsorbent | Experimental Conditions | $Q_{max}$ (mg/g) | Ref. |
|---|---|---|---|
| MIL-53(Al) metal-organic framework | pH: 7.5, adsorbent dosage: 0.1 g/L, contact time: 12 h, T: 303 K, initial concentration: 50 mg/L | 758.5 | [22] |
| NH$_4$Cl-induced activated carbon | pH: 6, adsorbent dosage: 0.8 g/L, contact time: 30 min, T: 303 K, initial concentration: 100 mg/L | 437 | [23] |
| Activated carbon/iron nanoparticles | pH: 5, adsorbent dosage: 1.5 g/L, contact time: 30 min, T: 298 K, initial concentration: 10 mg/L | 40.282 | [24] |
| Functional CoFe$_2$O$_4$-modified biochar | pH: 7, adsorbent dosage: 0.05 g/L, contact time: 60 min, T: 298 K, initial concentration: 50 mg/L | 99.99 | [25] |
| Activated carbon | pH: 5.5, adsorbent dosage: 0.06 g/L, contact time: 240, T: 298 K, initial concentration: 100 mg/L | 4.4 | [42] |
| Mg-Al layered double hydroxide/cellulose nanocomposite | pH: 7, adsorbent dosage: 0.1 g /L, contact time: 20 min, T: 298 K, initial concentration: 120 mg/L | 138.3 | [43] |
| Concrete-based hydrotalcite | pH: 5, adsorbent dosage: 0.1 g/L, contact time: 12 h, T: 298 K, initial concentration: 400 mg/L | 49.7 | [44] |
| Chitosan-coated Fe$_3$O$_4$@Cd-MOF microspheres | pH: 8, adsorbent dosage: 0.05 g/L, contact time: 240 min, T: 298 K, initial concentration: 50 mg/L | 103.09 | [45] |
| Fe$_3$O$_4$/SiO$_2$/CTAB–SiO$_2$ | pH: 5, adsorbent dosage: 0.009 g/L, contact time: 60 min, T: 298 K, initial concentration: 25 mg/L | 362.66 | [46] |
| Fe$_3$O$_4$@activated carbon | pH: 6, adsorbent dosage: 0. 5 g/L, contact time: 90 min, T: 298 K, initial concentration: 200 mg/L | 238.1 | [47] |
| CMT-g-PANI@Fe$_2$O$_3$ | pH: 7, adsorbent dosage: 0.005 g/L, contact time: 20 min, T: 298 K, initial concentration: 400 mg/L | 909.09 | Current work |

### 3.4. Adsorption Kinetic

For discovering and evaluating possible adsorption routes, equilibrium time, and adsorption rate-limiting phase of amoxicillin adsorbed over the CMT-g-PANI@Fe$_2$O$_3$ adsorbent used the PFO and PSO kinetics models. In addition, kinetic models are employed to define the adsorption process mechanism, for example, surface and chemical adsorption. The PFO is based on the adsorbent capacity and employed once adsorption happens via diffusion in a border layer, whereas the PSO expresses that the chemical adsorption procedure is the main controlling procedure. Figure 8A,B and Table 4 exhibit kinetic linear plots, and their calculated parameters. According to R$^2$, the difference between the estimated Q$_e$ and observed Q$_e$ values, the PSO is more appropriate for considering the *amoxicillin* adsorption kinetics on the CMT-g-PANI@Fe$_2$O$_3$. Consequently, chemical adsorption is the main mechanism in the adsorption of *amoxicillin* on the adsorbent and it displays that, in addition to *amoxicillin* molecules, the CMT-g-PANI@Fe$_2$O$_3$ adsorbent is involved in the adsorption procedure.

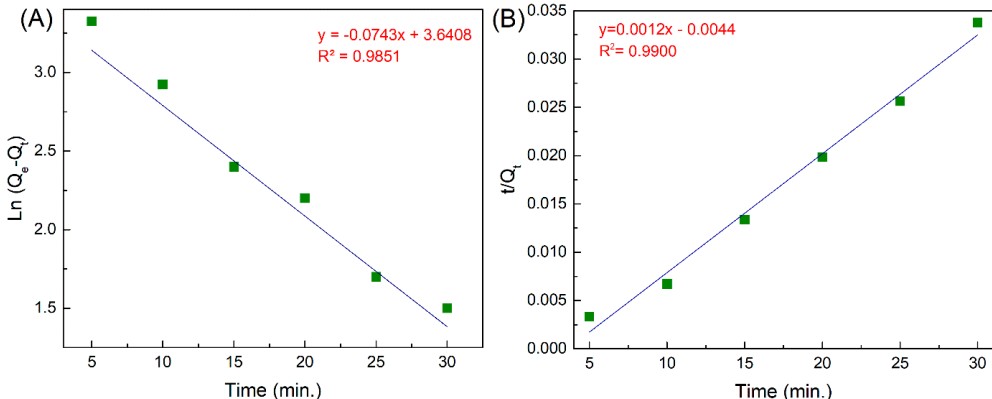

**Figure 8.** (**A**) Pseudo-first-order and (**B**) pseudo-second-order models (Contact time (5–30 min), pH 7, biosorbent dosage = 0.005 g, *amoxicillin* concentration = 400 mg/L, T = 298 K)).

**Table 4.** Kinetic parameters, for *amoxicillin* adsorption on the CMT-g-PANI@Fe$_2$O$_3$.

| Models | Parameters | |
|---|---|---|
| Pseudo-first-order | $k_1$ | 0.0743 |
| | Q$_e$ calculated | 38.122 |
| | Q$_e$ experimental | 788.73 |
| | R$^2$ | 0.9851 |
| Pseudo-second-order | $k_2$ | 0.000001 |
| | Q$_e$ | 833.33 |
| | Q$_e$ experimental | 788.83 |
| | R$^2$ | 0.9900 |

### 3.5. Desorption and Reusability

The reusability of adsorbent is a significant factor to create the adsorption procedure economically. The reusability of adsorbent can be effective for its use in numerous consecutive cycles without considerable performance debility. In the present work, we evaluated the *amoxicillin* desorption and reusability of CMT-g-PANI@Fe$_2$O$_3$ bioadsorbent in three consecutive cycles, and an ethanolic solution was investigated. In this regard, *amoxicillin* adsorbed onto the CMT-g-PANI@Fe$_2$O$_3$ was performed in optimal conditions and was immersed into ethanol (10 mL) under stirring at room temperature for 1 h to desorb *amoxicillin*. Afterward, CMT-g-PANI@Fe$_2$O$_3$ was collected by a magnet, washed with distilled water, and then dried for successive adsorption/desorption processes. Then, the released *amoxicillin* amount in the elution medium was measured using a UV-visible spectrophotometer. Figure 9 demonstrates that the adsorption percentage decreased from

98.59 % to 94.03%, and the desorption percentage decreased from 94.41% to 92.001% after three consecutive cycles, these results show that the CMT-g-PANI@Fe$_2$O$_3$ could continue to remove *amoxicillin* after three consecutive cycles of adsorption/desorption process without considerably losing adsorption capacity.

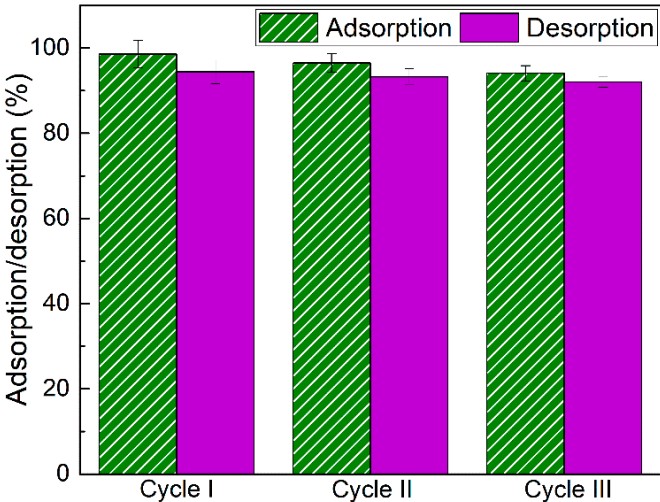

**Figure 9.** Desorption and reusability of the CMT-g-PANI@Fe$_2$O$_3$ bioadsorbent for *amoxicillin* adsorption.

### 3.6. Suggested Mechanism of Amoxicillin Adsorption

The pollutant adsorption mechanism by the adsorbent depends on the functional groups and morphology of the adsorbent, which can be created a synergistic effect in increasing the pollutant adsorption by the adsorbent. Considering that the CMT-g-PANI@Fe$_2$O$_3$ bioadsorbent has amine, carboxylate, and hydroxyl functional groups, it can create intermolecular interactions such as hydrogen bonding, electrostatic and π–π interactions with the *amoxicillin* antibiotic, and cause its efficient adsorption on the CMT-g-PANI@Fe$_2$O$_3$ bioadsorbent as shown in Figure 10.

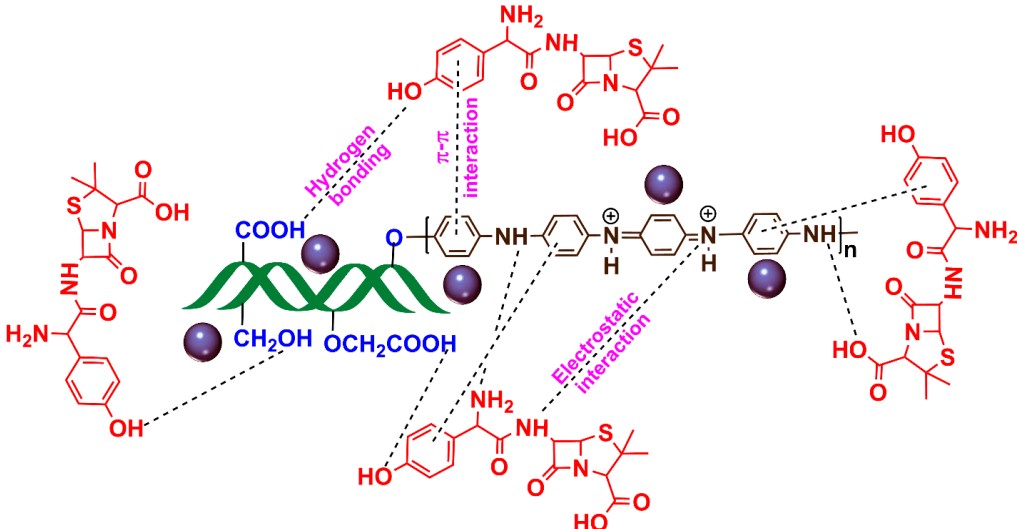

**Figure 10.** A schematic of the suggested mechanism of *amoxicillin* adsorption by CMT-g-PANI@Fe$_2$O$_3$ bioadsorbent.

### 4. Conclusions

A magnetic bionanocmposite was synthesized by an in situ copolymerization method from carboxy methyl tragacanth gum in the presence of aniline monomer and γ Fe$_2$O$_3$ nanoparticles (CMT-g-PANI@Fe$_2$O$_3$). The CMT-g-PANI@Fe$_2$O$_3$ bionanocmposite was

characterized by several analyses and employed as a bioadsorbent for the removal of *amoxicillin* antibiotic from contaminated water. A granular structure with good thermal stability (char yield 34 wt%), magnetization saturation (25 emu g$^{-1}$), and a specific surface area (8.0794 m$^2$/g) observed for CMT-g-PANI@Fe$_2$O$_3$ bionanocmposite. The maximum adsorption capacity (909.09 mg/g) was found at pH 7, an agitation time of 20 min, an adsorbent dose of 0.005 g, and an *amoxicillin* initial concentration of 400 mg/L. The Freundlich isotherm and pseudo-second-order kinetic models fit closely with the experimental data. The adsorption/desorption process showed that the CMT-g-PANI@Fe$_2$O$_3$ could continue to remove *amoxicillin* after three consecutive cycles without considerably losing adsorption capacity. The intermolecular interactions such as hydrogen bonding, electrostatic, and $\pi$–$\pi$ interactions between *amoxicillin* antibiotic and CMT-g-PANI@Fe$_2$O$_3$ were suggested for the adsorption mechanism of *amoxicillin* antibiotic by bionanocomposite.

**Author Contributions:** Conceptualization, E.N.Z.; methodology, S.S.M.; software, E.N.Z.; validation, E.N.Z., H.B. and M.T.; formal analysis, S.S.M.; investigation, E.N.Z., H.B. and M.T.; data curation, S.S.M.; writing—original draft preparation, E.N.Z.; writing—review and editing, E.N.Z., H.B. and M.T.; supervision, E.N.Z., H.B. and M.T.; project administration E.N.Z.; funding acquisition, S.S.M. All authors have read and agreed to the published version of the manuscript.

**Funding:** This research received no external funding.

**Data Availability Statement:** Data is available upon request.

**Acknowledgments:** The authors are grateful for the support of Damghan University's financial support.

**Conflicts of Interest:** The authors declare no conflict of interest.

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
