# Peer review of "Removal of Amoxicillin Antibiotic from Polluted Water by a Magnetic Bionanocomposite Based on Carboxymethyl Tragacanth Gum-Grafted-Polyaniline"

_water, doi:10.3390/w15010202_

Round 1

Reviewer 1 Report

The manuscript is, generally speaking, interesting and well presented, but some corrections are requested, e.g. line 361- the Authors seem to have missed some word. 

It would also be interesting to know what is the Authors' opinion on the selectivity of this material - what other pollutants could be removed with it?

Author Response

The manuscript is, generally speaking, interesting and well presented, but some corrections are requested, e.g. line 361- the Authors seem to have missed some words.

Response: Many thanks for your positive idea about our manuscript. The whole manuscript has been rechecked with Grammarly and corrected. Some sections of the manuscript have been revised (Please see the "Track Changes" version of the manuscript).  In addition, the mentioned line has been revised as follows:

“On the other hand, the comparison of Qmax for amoxicillin to other adsorbents reported in recent years showed that the CMT-g-PANI@Fe2O3 biosorbent has a higher maximum adsorption capacity (909.09 mg. g) than other adsorbents.”

It would also be interesting to know what is the Authors' opinion on the selectivity of this material - what other pollutants could be removed with it?

Response: Thank you so much for your valuable comment. Response to this question needs further experiments with at least two antibiotic drugs with similar chemical structures, for example, penicillin and amoxicillin. In the current work, we have not examined the adsorption process in the presence of two or many antibiotic drugs. But, in another work that was submitted to another journal, we used nanocomposite based on modified chitosan as molecularly imprinted material for the selective removal of an antibiotic such as levofloxacin.

As you know, if any drug is used during the preparation of nanocomposite, the final nanocomposite can be used as a molecularly imprinted material for the selective removal of drugs from water. In this regard, we have already published a review paper entitled “Remediation of pharmaceuticals from contaminated water by molecularly imprinted polymers: a review” (Environ Chem Lett 20, 2629–2664 (2022). https://doi.org/10.1007/s10311-022-01439-4) that can be useful.

Reviewer 2 Report

In current study author synthesized the Bbionanocomposite based on Carboxymethyl Tragacanth Gum-Grafted-Polyaniline and applied for adsorption of amoxicillin in water. Author added almost all required characterization studies required for the adsorbent material. Although the material is not novel but presented in good manner. Before accepting the manuscript I have few questions those need to be address.

1.       In title use the word “Removal” instead of “Remediation”.

2.       In preparation of carboxymethyl tragacanth gum how did you get the exact volume and concentration of solvents and chemicals?

3.       In start of abstract correct this sentence “Removing antibiotics from contaminated water” as “Removal of antibiotics from contaminated water”

4.       With the name of amoxicillin in italic form or name of any other antibiotic.

5.       In Figure 3 revisit the elemental composition in EDX graphs as few more peaks can be seen but missing in tables and discussion. How FESEM study is helpful for adsorption of amoxicillin on adsorbent?

6.       How paramagnetic character is important in adsorption process?

7.       What is the key point of magnetic oxide apart from adsorption?

8.       In Figure 6 how did you optimize temperature at 298 K? Temperature optimization graph is missing.

9.       Move the information about the isotherm models and kinetics equations into material and methods section.

10.   Discussion about the isotherm models and kinetics are not sufficient, give in details why adsorption mechanism follows Freundlich and PSO models?

11.   On the basis of isotherm model and kinetic study suggest the adsorption phenomena whether physical or chemical.

12.   In Table 3 add few adsorbents based on magnetic oxide.

13.   Page 12 line 414, correct this sentence “could continue to amoxicillin remove after” as “could continue to remove amoxicillin after”.

14.   Use the word synthesized instead of fabricated in many places in manuscript.

15.   Remove the word summery in start of conclusions.  

Reviewer 3 Report

Mosavi et al. demonstrated the adsorption of amoxicillin from water. A bionanocomposite comprising carboxymethyl tragacanth gum-grafted-polyaniline and γFe2O3 (CMT-g- 13

PANI@Fe2O3) was prepared by an in-situ copolymerization method. The adsorbent showed the maximum adsorption capacity (909.09 mg/g) at pH7, agitation time 20 minutes, adsorbent dose 0.005 g, and amoxicillin initial concentration 400 mg/L. The experimental data fitted well with the Freundlich isotherm and pseudo-second-order kinetic model. However, the manuscript needs minor revision for publication consideration. My specific comments are detailed below:

1.      Avoid abbreviations in the abstract.

2.      Provide a brief discussion of the literature-reported studies on the adsorptive removal of amoxicillin and/or analogues. This is important to understand the novelty and significance of this study.

3.      There is a difference between the adsorbent’s recoverability and reusability. It is suggested to rewrite this sub-section.

4.      How did the authors measure the amoxicillin concentration in samples using a UV-visible spectrophotometer? Also, provide a suitable reference for the same.

5.      Error bars are not clearly visible in the figures. Make suitable corrections.

6.      English editing is required to polish the manuscript. There are some typos as well.

7.      Table 3: It is essential to include the experimental conditions to present a clear picture of the adsorbent performance.

8.       The abstract should include the following points: a summary of your findings; new concepts and innovations demonstrated; a brief restatement of your hypotheses; a comparison with literature-reported results, and possible future work.

Round 2

Reviewer 3 Report

The authors have addressed the comments carefully and accordingly revised the manuscript. Therefore, it can be accepted for publication.